# Vertebral Heart Score and Vertebral Left Atrial Size as Radiographic Measurements for Cardiac Size in Dogs—A Literature Review

**DOI:** 10.3390/ani15050683

**Published:** 2025-02-26

**Authors:** Radu Andrei Baisan, Vasile Vulpe

**Affiliations:** Faculty of Veterinary Medicine, “Ion Ionescu de la Brad” Iasi University of Life Sciences, 700489 Iasi, Romania

**Keywords:** canine, cardiomegaly, left atrial enlargement, myxomatous mitral valve disease, thoracic radiography, VHS, VLAS

## Abstract

Cardiac size measurement is an important assessment tool for detection of cardiomegaly in dogs. Two of the most commonly used measurements are represented by vertebral heart score and vertebral left atrial size. Therefore, many research articles are available in the literature providing general or breed-specific reference ranges, as well as values in different heart disease or extra-cardiac conditions. This review aims to systematize the available data with regard to the reference ranges and cutoffs and cover the current use of these measurements in healthy and diseased dogs, based on the available literature.

## 1. Introduction

There are five basic methods used on a regular basis at the hand of a veterinary cardiologist, consisting of a specialized physical examination, electrocardiography, echography, biomarkers and radiology [1]. Cardiovascular studies led by Robert L. Hamlin and C. Roger Smith at Ohio State University began around 1960, pioneering studies in cardiac radiology [2,3]. The Caspary Institute of Animal Medical Center, established in 1961, provided facilities and support for cardiology research as well as notable contributions [4].

Nowadays, radiography of the thorax remains one of the most important assessments for heart enlargement and diagnosing left-sided congestive heart failure (CHF) in dogs. However, a radiologic report includes subjective and descriptive assessment of the organs and lesions and, whenever possible, measurements for quantifying the severity of a lesion. Having repeated measurements and comparing with previous measurements allows objective evidence of a condition progression or the effect of treatment [5]. Objective assessment of the heart size in dogs has been of interest for both researchers and clinicians for a long time. There have been a multitude of heart size measurements on radiography described in the past 30 years as radiology became more and more available and researchers developed different ways of assessing the heart size by describing new methods or modifying already available methods.

These measurements include vertebral heart size (VHS), vertebral left atrial size (VLAS), radiographic left atrial size (RLAD), heart to spine dimension, thoracic inlet heart size, the crossing line method, the manubrium heart score, the vertebral right heart index, modified VLAS (M-VLAS), the heart to single vertebra ratio, and the cardio-thoracic ratio [6,7,8,9,10,11,12,13,14,15,16]. Performing all these measurements in a patient is time-consuming, and different techniques may result in contradicting results, becoming confusing. In general, a measurement should be fast and easy to perform, should offer reliable values and, in accordance with gold standard methods, in this case echocardiography, should be certified by multiple studies, must be reproducible, and the operator’s experience should have a minimum influence on the results. Therefore, based on the previously mentioned criteria, this review will focus on two cardiac measurements, represented by VHS and VLAS. Since VHS is based on two axes of the heart (the long and the short axes) this measurement is able to assess the general cardiac size, while VLAS measures the size of the left atrium. Therefore, this review aims to cover the current use of two radiographic measurements of the heart in healthy and diseased dogs, based on the available literature.

## 2. Materials and Methods

The search for literature was conducted within the period 1995 and 2024 and included PubMed and Google Scholar databases. The search in the PubMed database was performed by the primary author using the “advanced search” option based on combined keywords, as follows: “((radiogr*) AND ((heart size) OR (cardiac size))) AND ((dog) OR (canine)”, resulting in 176 documents, “(((radio*) AND ((cardiac silhouette) OR (heart silhouette))) AND (vertebral)) AND ((dog) OR (canine))”, resulting in 21 documents, “((vertebral heart) OR (vertebral left atrial)) AND ((dog) OR (canine))”, resulting in 187 documents, and “(((radio*) AND ((heart) OR (cardiac))) AND ((dog) OR (canine))) AND (((artificial intelligence) OR (deep learning)) OR (machine learning))”, resulting in 27 papers. The papers were then screened by the primary author to eliminate any articles that did not fulfill the inclusion criteria. From the PubMed database, 68 papers were selected in this review. The papers were exported to EndNote v. 21, and duplicated documents were manually deleted. Four documents were retrieved from Google Scholar, and 13 documents were retrieved from other books’ and articles’ bibliographies. Articles in English that focused on veterinary radiographic measurement of heart size in dogs were selected. Articles that focused on other species, conference abstracts, and studies lacking full-text availability were excluded. The data from selected studies were extracted, synthetized, and organized according to the main subchapters of this review.

## 3. Measuring Method

### 3.1. Vertebral Heart Score Measurement

One of the most used quantitative methods of measuring the heart was described by Buchanan and Bucheler in 1995 [6]. This method proposed a measurement of the long and short axes of the heart on a lateral radiograph and comparing the dimension of the two lengths to the vertebral bodies. In brief, the cardiac long axis was measured from the ventral border of the left mainstem bronchus to the cardiac apex. The maximal short axis of the heart was measured in the central third region perpendicular to the long axis. The two measurements were then repositioned over the thoracic vertebrae, beginning with the cranial edge of the fourth thoracic vertebra (T4). The distance to the caudal caliper point was estimated to the nearest 0.1 vertebra. The short and long axes’ dimensions were then added to provide a vertebral heart sum, which represented an indicator of the cardiac size in relationship to the body length. The overall size of the heart was thus expressed as total units of vertebral length to the nearest 0.1 vertebra and termed the vertebral heart size [6]. This study also provided a variant of VHS measured on a VD/DV image; however, nowadays, it is less used and therefore will not be discussed in this review. Later, in 2005, Hansson et al. proposed a change in VHS measurement by redefining the short axis position such that the caudal reference point for the width was halfway between the dorsal and ventral border of the caudal vena cava [17]. Since then, many studies have used this modified technique.

### 3.2. Vertebral Left Atrial Size Measurement

Another measurement, VLAS, was described in 2018. In brief, it is performed on a lateral view of the thorax by drawing a line from the center of the most ventral aspect of the carina to the most caudal aspect of the left atrium where it intersects with the dorsal border of the caudal vena cava. Similar to VHS, a second line, equal in length to the first is drawn, beginning with the cranial edge of T4 and extending caudally just ventral and parallel to the vertebral canal. The VLAS is defined as the length of the second line expressed in vertebral body units to the nearest 0.1 vertebra [7]. The initial study was performed in 15 healthy dogs and 88 dogs with different stages of myxomatous mitral valve disease (MMVD). The median, upper, and lower quartile values for VLAS in healthy dogs proposed were 2.1 v. (1.8–2.3) and did not differ between the left and right lateral views [7].

### 3.3. Measurement Variations in Normal Subjects

Regarding VHS, the initial study performed on 100 normal dogs of various breeds and sizes, comprising dogs from 2 to 75 kg proposed reference values of 9.7 ± 0.5 vertebrae (v.) with a range between 8.5–10.6 v. regardless of the size, breed, sex or lateral recumbence [6]. However, subsequent studies have found differences in VHS reference ranges between different breeds and between left (LL) and right (RL) lateral recumbence. One study assessed the differences in VHS between LL and RL recumbences in 63 dogs of different size and breeds, resulting in a significantly higher VHS for RL (M ± SD 9.8 ± 0.6 v.) compared to LL (M ± SD 9.5 ± 0.8 v.); however, gender and dog size did not significantly differ [18]. Similarly, another study performed on 19 Beagle dogs obtained a significantly higher mean VHS in RL (10.5 v.) compared to LL recumbence (10.2 v). Also, studies assessing the effect of lateral recumbence for VHS in Indian Spitz, Labrador Retrievers, and mongrel dogs, as well as in common breeds in Iran supported the finding that the values of VHS in RL were larger compared to those obtained from LL [19,20]. On the other hand, Bagardi et al. (2022) found no significant difference in VHS in 30 Cavalier King Charles Spaniel dogs between left and right lateral recumbence [21]. Similarly, another study performed on 30 Chihuahua dogs did not find significant differences for VHS between left and right recumbence [13]. The effect of dog positioning on VHS remains unclear since there are contradicting results; however, within a breed, it seems that there is no major influence. However, whenever reference ranges are not available for both the left and right lateral positions, the clinician should take the view as where reference ranges are available. On the other hand, most of the studies have been performed at least in RL recumbence for heart analysis. Therefore, for consistency, we recommend taking the RL radiography of the chest when both lateral views cannot be performed. Regarding the effect of cardiac phase over VHS using fluoroscopic images, one study found a significant difference of 0.36 ± 0.14 vertebral units between systole and diastole, with higher values in the latter [22]. Furthermore, the VHS values were compared between conventional radiography and a computed tomography (CT) section. Although there was a high degree of correlation comparing the CT to the radiographic VHS, the authors explained that these measurements were not exactly the same since radiographs act as a two-dimensional representation of a three-dimensional structure, and choosing a single sagittal image on a CT would not truly represent the entire volume of the heart [23]. Moreover, it was shown that the respiratory phase, such as inspiration versus expiration, had no influence on the value of VHS in both healthy dogs and dogs with mitral regurgitation [24].

Several studies have evaluated the influence of body condition score (BCS) of the dogs over VHS. Bodh et al. in 2016 found no difference in VHS between different BCSs in Indian Spitz, Labrador Retrievers, and mongrel dogs [20]. Similarly, BCS did not show any correlation with VHS in healthy Chihuahua dogs [25]. On the other hand, it was found that Norwich Terriers with a BCS above 6 had a significantly greater VHS compared to those with a BCS < 5 [26]. Finally, one study found that BCS does not affect the reliability of VHS assessment in overweight dogs when performed by veterinarians with experience [27].

The initial study describing VLAS measurement reported no difference between left and right lateral views. However, a more recent study assessing the influence of radiographic recumbence in healthy Cavalier King Charles Spaniel dogs found that VLAS was significantly higher in LL recumbence (M ± SD 1.99 ± 0.25 v.) compared to RL (M ± SD 1.79 ± 0.3 v.) [21]. Differently from VHS, the cardiac cycle phase assessed by fluoroscopic images was found to have no influence on the results of the VLAS measurement [22]. Also, the respiratory phase did not show any influence on VLAS values in healthy dogs; however, the values were lower in the expiratory than in the inspiratory phase in dogs with mitral valve regurgitation, although the difference was very small (mean of 2.48 v. during inspiration versus a mean of 2.46 v. during expiration) [24]. Also, there was no influence of BCS and body weight on VLAS measurement [25].

## 4. The Use of VHS and VLAS in Clinical Practice

### 4.1. The Use of Reference Ranges for VHS and VLAS

Radiographic measurements of the heart can be used to assess cardiomegaly during a dog’s examination or by periodical examinations of the same animal. In the first case, there are also several ways of using these measurements. When breed-specific reference ranges are available, it is recommended to compare these with specific reported values, rather than using multi-breed studies. The available breed-specific reference ranges for VHS and VLAS are presented in Table 1 and Table 2. Moreover, some studies have reported cutoff values for the diagnosis of cardiomegaly or left atrial enlargement. The American College of Veterinary Internal Medicine (ACVIM) consensus guidelines for the diagnosis and treatment of myxomatous mitral valve disease proposed a cutoff of ≥11.5 v. for VHS or a comparable breed-adjusted VHS for diagnosing cardiomegaly in the absence of echocardiographic measurements in dogs with MMVD [28]. Also, Malcolm et al. proposed a cutoff of ≥2.5 v. for VLAS for detecting echocardiographic left atrial (LA) enlargement, defined as a left atrium to aorta ratio ≥ 1.6, with a sensitivity of 67% and a specificity of 84% [7]. However, similar to VHS, where breed-specific reference ranges are available, these should be used over multi-breed reference ranges.

**Table 1 animals-15-00683-t001:** Various breeds and breed-specific reference values published for vertebral heart score measurement in healthy dogs.

Breed	Recumbency	No. of Dogs	Data Expression	Reference Values	Reference
Various breeds	RL/LL	100 dogs	M ± SD (range)	9.7 ± 0.5 (8.5–10.6)	Buchanan and Bucheler, 1995 [6]
Various breeds	LL	11 young growing puppies	M ± SD	3 MO: 10.0 ± 0.52	Sleeper and Buchanan, 2001 [29]
				6 MO: 9.8 ± 0.42	
				12 MO: 9.9 ± 0.60	
				36 MO: 10.3 ± 0.58	
Various breeds	RL	80 dogs	Median (range)	10 (9.2–11)	Vezzosi et al., 2020 [30]
Small and large breed dogs	RL	120 dogs	M ± SD (95% CI)	Large breed dogs: 10.7 ± 0.5 (10.6–10.9)	Mostafa and Berry, 2017 [31]
				Small breed dogs: 10.3 ± 0.8 (10.1–10.5)	
Mongrel dogs	RL/LL	20 dogs	M ± SD	RL: 9.82 ± 0.21 LL: 9.62 ± 0.25	Bodh et al., 2016 [20]
Large breed dogs	RL/LL	56 dogs	M ± SD (range)	RL: 9.7 ± 0.59 (8.6–11.1)	Ghadiri et al., 2010 [19]
				LL: 9.6 ± 0.56 (8.6–10.9)	
Miniature Schnauzer	RL	272 dogs	95% reference interval	9.7–12.1	Murphy et al., 2024 [32]
Jack Russel Terrier	RL	302 dogs	Median (Q1–Q3)	10.8 (10.2–11)	Battinelli et al., 2024 [33]
Miniature Pinscher	RL	238 dogs	Median (Q1–Q3)	11.0 (10.3–11.2)	Battinelli et al., 2024 [33]
Brussel Griffon	RL	132 dogs	Median (Q1–Q3)	10.8 (10.2–11.0)	Battinelli et al., [33]
Shih-Tzu	RL	30 dogs	M ± SD	9.5 ± 0.6	Jepsen-Grant et al., 2013 [34]
Chihuahua	RL	10 dogs	M ± SD (95% CI)	9.66 ± 0.36 (9.4–9.92)	Ito, 2022 [35]
Chihuahua	RL/LL	30 dogs	M ± SD (95% CI)	RL: 10.0 ± 0.6 (8.9–11.0)	Puccinelli et al., 2021 [25]
				LL: 9.9 ± 0.3 (9.3–10.5)	
Poodle	RL/LL	40 dogs	M ± SD	9.72 ± 0.73	Azevedo et al., 2016 [36]
Poodle	RL	30 dogs	M ± SD (range)	10.12 ± 0.51 (9.2–11.1)	Pinto and Banon, 2010 [37]
Maltese	RL	81 dogs	M ± SD (95% CI)	9.53 ± 0.46 (9.4–9.6)	Baisan and Vulpe, 2022 [38]
Cavalier King Charles Spaniel	RL/LL	30 dogs	M ± SD (95% CI)	RL: 10.08 ± 0.56 (9.87–10.29)	Bagardi et al., 2022 [21]
				LL: 10.00 ± 0.41 (9.85–10.17)	
Cavalier King Charles Spaniel	RL	20 dogs	M ± SD (90% reference interval)	10.6 ± 0.5 (9.9–11.7)	Lamb et al., 2001 [39]
Yorkshire Terrier	RL	22 dogs	M ± SD (90% reference interval)	9.7 ± 0.5 (9.0–10.5)	Lamb et al., 2001 [39]
Pomeranian	RL	18 dogs	M ± SD	10.5 ± 0.9	Jepsen-Grant et al., 2013 [34]
Norwich Terrier	RL	31 dogs	M ± SD median (range)	10.5 ± 0.5	Taylor et al., 2020 [26]
				10.4 (9.4–11.6)	
Spitz	RL/LL	20 dogs	M ± SD	RL: 10.21 ± 0.13	Bodh et al., 2016 [20]
				LL: 10.03 ± 0.11	
Lhasa Apso	RL	18 dogs	M ± SD	9.6 ± 0.8	Jepsen-Grant et al., 2013 [34]
French and English bulldogs	RL	30 dogs	M ± SD	12.7 ± 1.7	Jepsen-Grant et al., 2013 [34]
Boston Terrier	RL	19 dogs	M ± SD	11.7 ± 1.4	Jepsen-Grant et al., 2013 [34]
American Pit Bull Terrier	RL	20 dogs	M ± SD	10.97 ± 0.33	Cardoso et al., 2011 [40]
Australian Cattle Dog	RL/LL	20 dogs	M ± SD (95% CI)	RL: 10.5 ± 0.5 (9.8–11.3)	Luciani et al., 2019 [41]
				LL: 10.3 ± 0.5 (9.9–11.3)	
Beagle	RL/LL	19 dogs	M ± SD (range)	RL: 10.5 ± 0.4 (9.8–11.2)	Kraetschmer et al., 2008 [42]
				LL: 10.2 ± 0.2 (9.2–10.7)	
Boxer	RL	20 dogs	M ± SD (90% reference interval)	11.6 ± 0.8 (10.3–12.6)	Lamb et al., 2001 [39]
Labrador Retriever	RL	25 dogs	M ± SD (90% reference interval)	10.8 ± 0.6 (9.7–11.7)	Lamb et al., 2001 [39]
Labrador Retriever	RL/LL	20 dogs	M ± SD	RL: 10.39 ± 0.15	Bodh et al., 2016 [20]
				LL: 10.22 ± 0.20	
Doberman Pinscher	RL	20 dogs	M ± SD (90% reference interval)	10 ± 0.6 (9.0–10.8)	Lamb et al., 2001 [39]
Rottweiler	RL/LL	38 dogs	M ± SD	9.8 ± 0.1	Marin et al., 2007 [43]
German Shepherd	RL	100 dogs	M ± SD	9.8 ± 0.5	Torad and Hassan, 2014 [16]
German Shepherd	RL	20 dogs	M ± SD (90% reference interval)	9.7 ± 0.7 (8.7–11.2)	Lamb et al., 2001 [39]
Border Collie	RL/LL	12 dogs	M ± SD	RL: 10.10 ± 0.51	Prado et al. 2024 [44]
				LL: 9.95 ± 0.47	
Whippets	RL/LL	44 dogs	M ± SD	RL: 11.0 ± 0.5	Bavegems et al., 2005 [45]
				LL: 11.3 ± 0.5	
Greyhounds	RL/LL	42 dogs	M ± SD	10.5 ± 0.1	Marin et al., 2007 [43]

RL—right lateral; LL—left lateral; M ± SD—mean and standard deviation; Q1–Q3—quartile 1–quartile 3; CI—confidence interval; MO—months old.

### 4.2. The Use of Serial Measurements for VHS and VLAS in Disease Progression

The use of serial measurements in the same dog has several advantages, such as eliminating inter-individual variation, and the proof of a good predictor for CHF has been demonstrated by several studies. One study assessed the rate of increase in VHS prior to CHF in Cavalier King Charles Spaniel dogs, showing that VHS steadily increased until the last interval (8.6 months), when the values were the highest. Moreover, the rate of change per month did not increase until the last interval, which ended with the onset of CHF. An increase in >0.08 vertebral units per month was associated with CHF [46]. In another study, the rate of change in VHS increased little until the last year before CHF in dogs with MMVD [47]. Similarly, in a group of dogs with preclinical MMVD with cardiomegaly, VHS increased in the last 12 months prior to CHF, while in those that did not experience CHF, the values remained stable. The maximal rate of change was observed in the period immediately before the onset of CHF [48]. The potential for risk assessment in dogs with preclinical MMVD has also been evaluated for VLAS. One study found that the VLAS values of dogs that developed CHF was significantly higher compared to the dogs that remained preclinical, suggesting that VLAS may be useful in identifying the risk of CHF in dogs with MMVD within the next 180 days. Moreover, a VLAS cutoff of >2.95 v. had a specificity of 85.7% for identifying dogs progressing to CHF; however, the sensitivity was only 40% [49]. Furthermore, a monthly rate of change in VLAS of 0.02 was associated with the probability of developing CHF in dogs with MMVD [49].

**Table 2 animals-15-00683-t002:** Various breeds and breed-specific reference values published for vertebral left atrial size measurement in healthy dogs.

Breed	Recumbency	No. of Dogs	Data Expression	Reference Values	Reference
Various breeds	RL	36 dogs	M ± SD (range)	1.79 ± 0.29 (1.69–1.89)	Marbella Fernandez and Montoya-Alonso, 2023 [50]
Various breeds	RL/LL	15 dogs	Median (Q1–Q3)	2.1 (1.8–2.3)	Malcolm et al., 2018 [7]
Various breeds	RL	80 dogs	Median (range)	1.9 (1.3–2.2)	Vezzosi et al., 2020 [30]
Miniature Schnauzer	RL	272 dogs	95% reference interval	1.7–2.4	Murphy et al., 2024 [32]
Jack Russel Terrier	RL	302 dogs	Median (Q1–Q3)	2.1 (2–2.2)	Battinelli et al., 2024 [33]
Miniature Pinscher	RL	238 dogs	Median (Q1–Q3)	2 (1.8–2.1)	Battinelli et al., 2024 [33]
Brussel Griffon	RL	132 dogs	Median (Q1–Q3)	2 (1.9–2.1)	Battinelli et al., 2024 [33]
Cavalier King Charles Spaniel	RL/LL	30 dogs	M ± SD (95% range)	RL: 1.79 ± 0.3 (1.68–1.90)	Bagardi et al., 2022 [21]
				LL: 1.99 ± 0.25 (1.90–2.09)	
Maltese	RL	81 dogs	Median (Q1–Q3) 95% CI	2 (1.8–2.1)	Baisan and Vulpe, 2022 [38]
				1.5–2.3	
Chihuahua	RL	30 dogs	M ± SD (95% CI)	1.8 ± 0.2 (1.3–2.1)	Puccinelli et al., 2021 [25]
Pug	RL	32 dogs	Median (95% reference interval)	2 (1.1–2.8)	Wiegel et al., 2022 [51]

RL—right lateral; LL—left lateral; M ± SD—mean and standard deviation; Q1–Q3—quartile 1–quartile 3; CI—confidence interval.

## 5. Vertebral Heart Score and Vertebral Left Atrial Size in Heart Disease

### 5.1. Vertebral Heart Score in Dogs with Heart Disease

The vertebral heart score has been shown to increase with experimental cardiomegaly induced by high-rate pacing in dogs [52]. Dogs with different preclinical cardiac diseases, such as dilated cardiomyopathy (DCM) (median and Q1–Q3 of 11 v; 10.6–11.8) and MMVD (median and Q1–Q3 of 11.1 v; 10.6–11.7), have been shown to have a higher VHS compared to normal dogs (median and Q1–Q3 10.6 v; 10.1–11.1). Also, dogs with clinical DCM (median and Q1–Q3 of 11.9 v; 11.2–12.9) and those with clinical MMVD (median and Q1–Q3 of 12.5 v; 11.7–13) had higher values compared to preclinical dogs [53]. One study assessed the values of VHS in a large population of 1358 dogs with MMVD, providing ranges for VHS in dogs of different classes of severity. In this study, the ranges (minimum–maximum) for dogs in MMVD class B1 were 8–13.3 v, with a median of 10.5 v, for B2 were 8.9–13.5 v. with a median of 11.4 v, and for class C were 8.8–14.5 v, with a median of 12 v [54]. Although the limits were very wide, and there is a major overlap between the groups, there was a significant difference for VHS between all three groups of dogs. Furthermore, a VHS value of 11 v. had a sensitivity of 66% and a specificity of 92% for differentiating between MMVD dogs with (B2) and without (B1) cardiomegaly, while a VHS cutoff of 12 v. had a sensitivity of 60% and a specificity of 86% in differentiating between dogs in class B2 and class C [55]. Another study performed on dogs with different classes of MMVD reported a VHS value (M ± SD) of 10.28 ± 0.76 v. in class B1, 11.61 ± 0.93 v. in class B2, and 12.04 ± 1.52 v. in class C. Median and range (minimum–maximum) VHS values for dogs diagnosed with different classes of MMVD is depicted in Figure 1A-C. Moreover, similar to the previous study [55], a VHS cutoff ≥ 11.1 v. had a sensitivity of 78% and a specificity of 96% in differentiating dogs with left atrial enlargement compared to those with a normal LA [14].

Of great interest was the ability of VHS to discriminate between MMVD dogs with and without cardiomegaly based on ACVIM classification (Figure 2) [28]. However, it should be noted that ACVIM class B1 is represented by dogs without echocardiographic evidence of cardiomegaly or with LA or LV enlargement, while B2 dogs must have both echocardiographic LA and LV enlargement [28]. Therefore, it is understood that when assessing radiographic cardiomegaly based on ACVIM consensus classification, some dogs with either LA or LV are included in the group of dogs without cardiomegaly. Nevertheless, evaluation of heart size should be regarded in terms of prognosis and severity rather than comparing the radiographic heart size with echocardiographic measurements. One study investigating the VHS value in differentiating between MMVD dogs with and without cardiomegaly reported a median (Q1–Q3) of 10.8 v. (8.8–12.8) in B1 dogs and 11.5 v. (10–13.5) in B2 dogs, with a significant difference in VHS values between the two groups. A cutoff of 12 v. had a sensitivity of 37% and a specificity of 99% for differentiating between dogs with and without cardiomegaly [56]. A similar study, investigating the VHS in dogs with MMVD reported a median (IQR) of 10.5 v. (0.74) in dogs without cardiomegaly and 11.6 v. (0.77) in dogs with cardiomegaly. Also, this study proposed a VHS cutoff of 10.8 v. for differentiating between dogs with and without cardiomegaly, with a sensitivity of 91.1% and a specificity of 69.5%, while a cutoff of 11.2 v. showed a lower sensitivity of 61.8% but a higher specificity, of 83.3% [60]. When echocardiographic LA enlargement was defined as LA/AoD (left atrium to aortic diameter in the long axis) > 2.54 and LA/Ao (left atrium to aortic diameter in the short axis) >1.68, the optimal VHS cutoff for differentiating between dogs with and without LA enlargement was 11.1 v, with a sensitivity of 75.8% and a specificity of 76% [61]. In addition to these data, the ACVIM consensus guidelines for the diagnosis and treatment of myxomatous mitral valve disease in dogs propose a VHS value > 10.5 v. with echocardiographic evidence of LV and LA enlargement as a criterion for diagnosing stage B2. However, in the absence of echocardiographic measurements, the breed-independent proposed VHS cutoff for a B2 class was 11.5 v [28]. Borgarelli et al. (2021) found that dogs with MMVD and a VHS value of > 11.3 v. was had a 2.56-fold higher risk of developing CHF [62]. When dogs with MMVD were classified according to their LA enlargement severity, regardless of the ACVIM class, a VHS cutoff of 10.7 v. had a sensitivity of 79% and a specificity of 97% in detecting LA enlargement, defined as LA/Ao > 1.6, while a cutoff of 11 v. had a sensitivity of 92% and a specificity of 90.7% in differentiating between mild (LA/Ao between 1.6 and 1.89) and moderate (LA/Ao between 1.9 and 2.2) LA enlargement. Furthermore, a VHS cutoff of 11.5 v. had a sensitivity of 90% and a specificity of 78% in detecting severe (LA/Ao > 2.2) LA enlargement [63]. In Cavalier King Charles Spaniel dogs diagnosed with MMVD, VHS increased with heart disease. The median VHS in class A was 10.1 v., in B1 it was 10.5 v., and in B2 it was 11.5 v. The cutoff value for differentiating between dogs with and without cardiomegaly was 11 v., with a sensitivity of 80% and a specificity of 74% [57]. In Chihuahua dogs, the VHS increased with the degree of LA and LV enlargement. A VHS cutoff > 10.5 v. detected dogs with an echocardiographic LA/Ao > 1.6 and a LVIDd_n (left ventricular diameter in diastole indexed to body-weight) > 1.7, with a sensitivity of 64% and a specificity of 93%. A VHS cutoff > 11.1 v. detected those with an echocardiographic LA/Ao > 2 and a LVIDD_n > 1.7 with a sensitivity of 82% and a specificity of 78% [35]. Also, in Brittany Spaniel dogs, the VHS was significantly larger in dogs with MMVD (M ± SD 11.9 ± 1.1 v.), with a range between 11.4 to 13.4 v, compared to control dogs (M ± SD 10.6 ± 0.2 v.) that had a VHS range between 10.4 to 10.8 v [64].

**Figure 2 animals-15-00683-f002:**
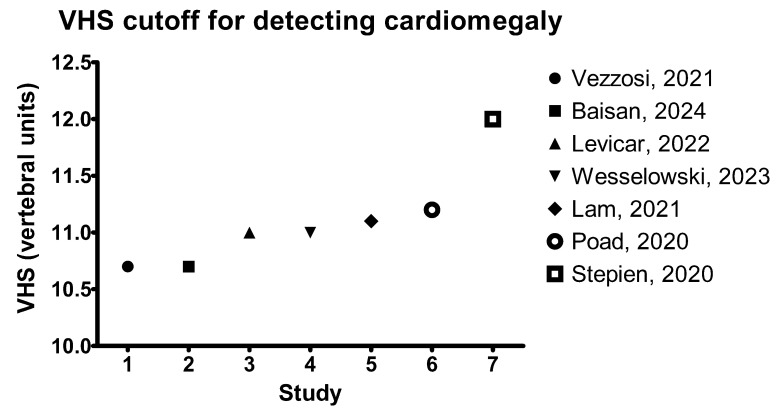
Graphic depicting the VHS cutoff values for discriminating dogs with cardiomegaly secondary to MMVD, based on different studies available in the literature [14,55,56,57,59,60,63].

Although there are many studies assessing the VHS ranges and proposing cutoff values for diagnosing cardiomegaly with acceptable sensitivity and specificity, a study performed on a large population of dogs with MMVD found that radiographic cardiomegaly was found in 33% of dogs with echocardiographic normal LA and LV. Moreover, 21% of dogs with mild to moderate LA enlargement and 14% of dogs with severe LA enlargement had a normal VHS [54]. This demonstrates that VHS as a single measurement should be carefully interpreted and, when possible, cardiomegaly should be confirmed by echocardiography.

The vertebral heart score was assessed for the potential of discriminating between dogs with cough from cardiac or non-cardiac reasons. Guglielmini et al. (2009) reported that dogs with a non-cardiac cough had a significantly lower VHS (M ± SD 11 ± 0.9 v.) compared to dogs with cough secondary to a cardiac reason (M ± SD 12.8 ± 0.9 v.). Moreover, a cutoff value of 11.4 v. could discriminate between dogs with a cardiac-related and non-cardiac-related cough, with a sensitivity of 92% and a specificity of 75% [65]. Similarly, the VHS measurement has been assessed in dogs with cardiac and non-cardiac respiratory distress. As expected, the VHS was larger in dogs with CHF, with a median (Q1–Q3) of 12.9 v. (11.9–14.1) compared to dogs with non-cardiac respiratory distress, with a median (Q1–Q3) of 11.1 v. (10.1–11.8) [66].

Dogs with heartworm disease (HWD) had a larger VHS (M ± SD 10.74 ± 1.12 v.) compared to the initial reference ranges reported by Buchanan and Bucheler in 1995 [6,67]; however, in this study, the authors did not use their own group of healthy dogs for comparison nor differentiate between the severity of the degrees of cardiomegaly secondary to HWD. Another study that evaluated changes in VHS during HWD treatment reported an increase in VHS over time (after 90 and 270 days respectively) despite therapy, probably as a consequence of chronic persistent pulmonary hypertension that ultimately leads to right-sided CHF. Moreover, dogs with pulmonary hypertension had larger values of VHS compared to those without pulmonary hypertension [68]. Also, the VHS was evaluated in dogs with pericardial effusion (PE) and compared to unilateral cardiac diseases, such as mitral regurgitation, tricuspid regurgitation, and aortic or pulmonic stenosis, and with bilateral cardiac disorders, such as DCM or bilateral atrio-ventricular valve regurgitation. The VHS in dogs with PE was larger (M ± SD 13.1 ± 1.1 v.) compared to both, unilateral (M ± SD 11.6 ± 1 v.), and bilateral (M ± SD 11.7 ± 1 v.) cardiac disorders. However, when dogs with PE were divided in mild, moderate, and severe, there was no significant difference in VHS values between these groups [69].

Lastly, the VHS method has been assessed in dogs before and after patent ductus arteriosus (PDA) occlusion. The VHS value before occlusion (M ± SD 11.7 ± 0.14 v.) was larger compared to the same dogs more than 12 months post-occlusion (M ± SD 10.7 ± 0.14 v.), demonstrating a normalization of the cardiac size once the procedure was performed [70]. Furthermore, VHS positively correlated with NT-pro BNP hormone in dogs with PDA [71].

### 5.2. Vertebral Left Atrial Size in Dogs with Heart Disease

In 2018, Malcolm et al. published a new radiographic measurement called VLAS, focusing on quantifying the left atrium [7]. This measurement was easy to use and since then has raised interest in the diagnostic potential of VLAS for diagnosing LA enlargement. Moreover, VLAS was immediately included in the ACVIM consensus guidelines for the diagnosis and treatment of myxomatous mitral valve disease in the protocol for diagnosing stage B2 [28], which increased even more the interest for evaluation of this measurement. In the initial study describing VLAS, the authors reported a positive correlation with the echocardiographic LA dimension. The reported median (Q1–Q3) for MMVD dogs in stage B1 was 2.1 v. (2.0–2.4), for B2 it was 2.6 v. (2.3–2.9) and for stages C and D it was 3.0 v. (2.7–3.6). The VLAS median and range (minimum–maximum) values for dogs with different classes of MMVD based on available publications are depicted in Figure 3A–C.

Furthermore, a VLAS cutoff > 2.5 v. had a sensitivity of 67% and a specificity of 84% for detecting dogs with echocardiographic LA enlargement [7]. Similarly, Mikawa et al. (2020) found a positive correlation between VLAS and echocardiographic LA/Ao measurement, while a VLAS cutoff of > 2.6 v. had a sensitivity of 95% and a specificity of 84% in differentiating MMVD dogs between stages B1 and B2 [58]. Another study assessing the diagnostic accuracy for detecting MMVD stage B2 reported a cutoff value of 2.5 v., with a sensitivity of 70% and a specificity of 84% [56], while Lam et al. 2021, reported a cutoff ≥2.4 v., with a sensitivity of 80.5% and a specificity of 96.6%. In a study performed on 100 dogs with MMVD, the VLAS cutoff for differentiating between dogs with and without cardiomegaly was also 2.5 v., with a sensitivity of 69% and a specificity of 97% [50]. Other authors have reported slightly different cutoff values. For example, a study performed on 70 preclinical dogs with MMVD reported a VLAS cutoff value of 2.3 v. for differentiating between preclinical dogs without and with cardiomegaly, with a sensitivity of 71.8% and a specificity of 77.4% [60]. Similarly, Duler et al. 2021, reported the same cutoff value (2.3 v.) for differentiating dogs without and with echocardiographic evidence of left atrial enlargement; however, the echocardiographic LA/Ao cutoff value for left atrial enlargement was 1.68. Moreover, this study concluded that VLAS was a superior indicator for left atrial enlargement compared to VHS [61]. A VLAS cutoff of 2.2 v. for detecting LA enlargement alone in dogs with MMVD was reported by Vezzosi et al. 2021 [59]. Furthermore, this study proposed a VLAS cutoff of 2.4 v. in discriminating between dogs with normal to mild and moderate to severe LA enlargement. However, when dogs were classified according to ACVIM staging, a VLAS cutoff > 2.4 v. was able to detect class B2 dogs, with a sensitivity of 66% and a specificity of 100% [59]. The VLAS cutoff values for diagnosing cardiomegaly in dogs with MMVD, based on available studies, are presented in Figure 4.

Interestingly, despite the similar reported cutoff values available in the literature, the ACVIM consensus guidelines for the diagnosis and treatment of myxomatous mitral valve disease proposed a cutoff value of > 3 v. for diagnosing MMVD stage B2 in the absence of echocardiography [28]. However, at the time of the ACVIM consensus publication (2019), only one study regarding VLAS was available, and one must be aware that this high cutoff value should be used whenever echocardiography is not available. One study reported the VLAS cutoff values between different classes of dogs with MMVD. A VLAS cutoff value of 2.3 v. was able to differentiate between B1 and B2 dogs with a sensitivity of 84% and a specificity of 73%, while a cutoff value of 2.7 v. discriminated between B2 and C classes, with a sensitivity of 76% and a specificity of 62% [55]. When VLAS was assessed in MMVD dogs with different degrees of echocardiographic LA enlargement, regardless of the ACVIM class, a cutoff > 2.5 v. detected dogs with left atrial enlargement (LA/Ao > 1.6), with a sensitivity of 73% and a specificity of 94%, while a cutoff of 2.7 v. differentiated between dogs with mild (LA/Ao between 1.6 and 1.89) and moderate (LA/Ao between 1.9 and 2.2) LA enlargement, with a sensitivity of 80% and a specificity of 92%. Furthermore, a VLAS cutoff > 2.9 v. was able to diagnose dogs with severe LA enlargement (LA/Ao > 2.2), with a sensitivity of 83% and a specificity of 86% [63]. In Cavalier King Charles Spaniel dogs with MMVD, the median (Q1–Q3) in class B1 was 1.8 v. (1.7–2.1), while in class B2 it was 2.3 v. (1.9–2.6). When the diagnostic accuracy for differentiating dogs without and with cardiomegaly was investigated, a cutoff value of 2.3 v. showed a sensitivity of 51.1% and a specificity of 92.3% [57]. Vertebral left atrial size was also evaluated in dogs with DCM. The median (Q1–Q3) value of VLAS for dogs with preclinical DCM was 2.2 v. (1.0–2.5), while for dogs with clinical DCM it was 2.5 v. (2.2–2.9) [53]. Ultimately, the use of VLAS for discriminating respiratory distress from cardiac and non-cardiac origin in dogs was also evaluated. The VLAS median (Q1–Q3) in dogs with respiratory distress with non-cardiac origin was 2.1 v. (1.9–2.2), while in dogs with respiratory distress from cardiac origin it was 3.0 v. (2.8–3.2). Compared to VHS, VLAS was a significantly more accurate predictor for CHF, with an optimal cutoff > 2.3 v. and a sensitivity of 93% and a specificity of 82.5% [66].

## 6. Repeatability and Reproducibility

A very important aspect regarding a measurement is the reproducibility and repeatability. These aspects have also been addressed in several studies regarding VHS and VLAS. A study performed by Vezzosi et al. (2020) on radiographs obtained from healthy dogs reported excellent agreement between observers with different experience, with an intraclass correlation coefficient (ICC) of 0.91 for VLAS and 0.95 for VHS. Moreover, the intra-observer agreement was also very high, with an ICC of 0.95 for VLAS and 0.94 for VHS [30]. Also, a study measuring cardiac size in dogs during a follow-up study reported that sequential paired comparison between two observers measuring VHS showed no decrease between four serial evaluations, indicating no learning effect in performing this measurement [46]. In a different study, two observers showed a 0.08 v. and 0.03 v. intra-observer variability, respectively, while the inter-observer analysis revealed a 0.26 v. difference between the two observers [8]. In healthy Maltese dogs, both VHS and VLAS showed excellent intra-observer agreement, with an ICC of 0.99 and 0.9, respectively, while the inter-observer agreement was 0.87 for VHS and 0.73 for VLAS [38]. Similarly, in Pug dogs, the intra-observer agreement between observers with different experience was almost perfect for both VHS and VLAS and also almost perfect in the inter-observer agreement for VHS. However, the inter-observer agreement for VLAS was only moderate, with an ICC of 0.49 [51]. When VHS was assessed in Greyhound dogs by multiple observers, there was no significant difference detected among observers with different levels of experience [43]. The respiratory phase did not show a major influence on the inter-observer variability when measuring VHS and VLAS. The ICC for VHS during inspiration was 0.93, while in expiration it was 0.90, and for VLAS it was 0.85 and 0.86 for inspiration and expiration, respectively [24]. Several studies have also assessed the reproducibility of the two radiographic measurements in dogs with heart disease. Levicar et al. (2018) found an almost perfect inter-observer agreement for VHS and good agreement for VLAS in healthy dogs and dogs with MMVD, while the intra-observer agreement was almost perfect for both VHS and VLAS [72]. Similarly, Poad et al. (2020) found an almost perfect agreement between two observers (ICC of 0.932) for measuring VLAS in dogs with preclinical MMVD [60]. Comparable results were also reported by Lam et al. (2021) in dogs with MMVD, where both intra and inter-observer analysis showed good to excellent agreement for both VHS and VLAS measurements [14]. Hansson et al. in 2005 assessed the inter-observer variability for VHS using observers with four different levels of experience, from the highest (European Diplomates in Veterinary Diagnostic Imaging) to the lowest (5th-year veterinary students). The coefficient of variation among all observers was between 2.0% and 2.6%. Moreover, the mean difference among all observers was 1.0 ± 0.3 v, with ranges between 0.5 v. and 1.9 v. This study concluded that VHS is independent of the observer’s experience; however, it seems dependent on the individual observer’s selection of the reference point [17]. A more recent study performed on normal dogs and dogs with MMVD and using multiple observers with different experience reported that there were differences in both VLAS and VHS between observers with more experience compared to those with less experience [73]. Data on within- and between-observer agreement obtained from different studies available in the literature are presented in Table 3. These studies demonstrate that both VHS and VLAS are reliable methods and have minimal dependence on the user’s experience. However, for achieving the best results, one must bear in mind that radiography should have a good quality, and artifacts must be minimized. Moreover, in some cases, the reference points could be difficult to assess even for the most experienced radiologists, and this remains a drawback when radiologic cardiac measurements are performed.

## 7. Future Perspectives

Recently, the development of computer-aided algorithms to support clinical diagnosis in both human and veterinary medicine has gained interest [74,75,76]. In humans, the development of artificial intelligence (AI) algorithms has been used for heart disease prediction and automatic image interpretation [77]. Moreover, a deep learning AI-based model for the diagnosis of mitral regurgitation from chest radiography has been developed [78]. Artificial intelligence use in veterinary radiology is in its early phase of development. However, some studies already have shown satisfactory results. Banzato et al. (2021) developed a method based on a multi-label deep convolutional neural network (CNN) for the classification of thoracic radiographs in dogs as unremarkable, cardiomegaly, alveolar pattern, bronchial pattern, interstitial pattern, mass, pleural effusion, pneumothorax, and megaesophagus. The method showed good results for all the included radiographic findings, except for bronchial and interstitial patterns [79]. In another study, 792 right lateral radiographs from canine patients with thoracic radiographs and contemporaneous echocardiograms were used to train, validate, and test a CNN algorithm for the detection of left atrial enlargement. The overall accuracy of the accuracy-driven CNN algorithm and veterinary radiologists was identical, while concordance between the two approaches was 85.19% [75]. Furthermore, the CNN algorithm applied on radiographs from dogs with MMVD showed good performance in determining the disease stage. This algorithm showed good accuracy in identifying dogs in ACVIM stages B1 and C + D, but moderate accuracy in classifying dogs at the B2 stage [80]. When dogs were classified according to ACVIM [28] staging and MINE [81] (Mitral INsufficiency Echocardiographic score), the developed algorithm showed good accuracy in predicting MMVD stages based on both classification systems, proving a potentially useful tool in the early diagnosis of canine MMVD [82]. When comparing AI-calculated VHS with three board-certified cardiologists’ measurements, the 95th percentile absolute difference was 1.05 vertebrae (95% confidence interval: 0.97 to 1.20 vertebrae) with a mean bias of −0.09 vertebrae (95% confidence interval: −0.12 to −0.05 vertebrae) [83]. In another study, the AI model was trained to automatically calculate an adjusted volume heart index calculated as the total area of the heart multiplied by the heart’s height and divided by the fourth thoracic vertebral body, and the authors concluded that the new method might outperform the current standard practice [84]. Furthermore, the potential of AI-based models in detecting pulmonary edema on radiography revealed an accuracy, sensitivity, and specificity of the AI-based software compared to the radiologist diagnosis of 92.3%, 91.3%, and 92.4%, respectively [85]. These results support the hypothesis that AI could assist with short-term decision-making when a radiologist is unavailable. To date, the available data from the literature show promising advantages from using AI technology in the medical field in the future. Not only can AI models process the information faster and offer quick response, but also it can be used whenever a specialist or experienced clinician is unavailable. Moreover, in the future, AI models could be built to include a larger amount of clinical data and guide the clinician towards a diagnosis or appropriate therapy within a shorter period of time. With regard to cardiac size measurements on radiography, breed-specific reference ranges could be used to build up AI models and offer an accurate result, eliminating user experience and human error when deciding the reference points for such techniques.

## 8. Conclusions

Many methods for quantification of the cardiac size on radiography are available in dogs, yet most studies focus on VHS and VLAS. To date, there has been a substantial amount of information regarding these measurements published in the veterinary literature. However, further studies on breed-specific ranges are still needed. Both VHS and VLAS are reliable, and the user’s experience has a low impact on the result. However, these methods seem dependent on the individual’s choice for selection of the reference points. Finally, radiographic measurements of the heart could add valuable clinical information, especially when echocardiography is not available, yet the results should be interpreted in a clinical setting.

## Figures and Tables

**Figure 1 animals-15-00683-f001:**
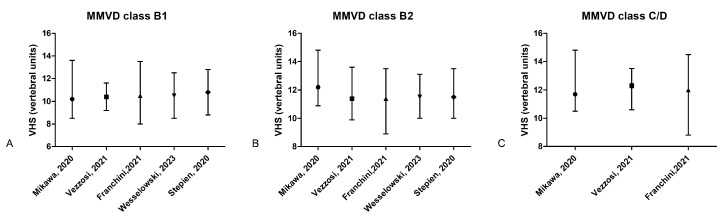
Graphics representing the VHS ranges and cutoffs based on the literature data: (**A**–**C**) bars representing the VHS median and range (minimum–maximum) for dogs diagnosed with MMVD class B1, B2, and C/D, according to available studies [54,56,57,58,59].

**Figure 3 animals-15-00683-f003:**
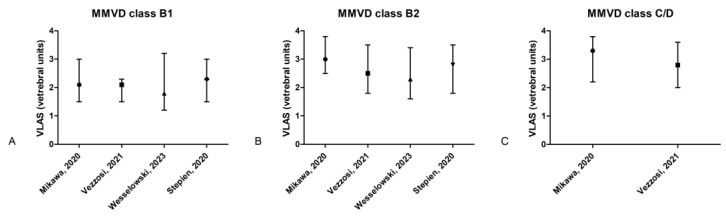
Graphics representing the VLAS ranges and cutoffs based on the literature data: (**A**–**C**) bars representing the VHS median and range (minimum–maximum) for dogs diagnosed with MMVD class B1, B2, and C/D, according to available studies [56,57,58,59].

**Figure 4 animals-15-00683-f004:**
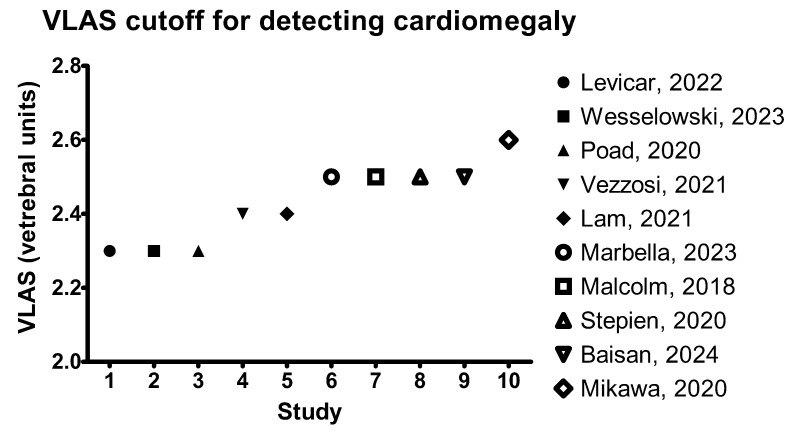
Graphic depicting the VLAS cutoff values for discriminating dogs with cardiomegaly secondary to MMVD, based on different studies available in the literature [7,14,50,55,56,57,58,59,60,63].

**Table 3 animals-15-00683-t003:** Inter-observer and intra-observer correlation coefficient for vertebral heart score (VHS) and vertebral left atrial size (VLAS) in dogs among different studies.

Study	No. of Observers	Measurement	ICC Between Observers	ICC Within Observers
Malcolm et al., 2018 [7]	4	VLAS	0.87	0.93
Baisan et al., 2022 [38]	3	VHS	0.87	0.99
		VLAS	0.73	0.9
Chhoey et al., 2020 [24]	2	VHS	0.93	N/A
		VLAS	0.85	N/A
Levicar et al., 2022 [55]	3	VHS	0.96	0.978–0.985
		VLAS	0.85	0.929–0.964
Poad et al., 2019 [60]	2	VLAS	0.932	N/A
Lam et al., 2021 [14]	3	VHS	0.94	0.99
		VLAS	0.92	0.97
Wiegel et al., 2022 [51]	3	VHS	0.89	0.98
		VLAS	0.49	0.91
Vezzosi et al., 2020 [30]	2	VHS	0.95	0.94
		VLAS	0.91	0.95
Lord et al., 2011 [46]	2	VHS	N/A	0.933–0.943

VHS—vertebral heart score; VLAS—vertebral left atrial size; ICC—intraclass correlation coefficient; N/A—not available.

## Data Availability

No new data were created or analyzed in this study.

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
