# Peer review of "Vertebral Heart Score and Vertebral Left Atrial Size as Radiographic Measurements for Cardiac Size in Dogs—A Literature Review"

_animals, 2025, doi:10.3390/ani15050683_

Round 1
Reviewer 1 Report
Comments and Suggestions for Authors
The great effort, dedication, and time invested by the authors in this work, as well as the valuable contribution it aims to make to the field of veterinary radiography, are truly appreciated. While this work provides interesting insights into the use of VHS and VLAS in veterinary radiography, including its evolution over time, in the reviewer’s opinion, the review does not present sufficiently novel or innovative findings to advance current knowledge in the field. Journals of this level prioritize groundbreaking contributions. A more comprehensive and critical evaluation of existing studies, with a clear identification of gaps and future research directions, would strengthen the contribution.
Author Response
Comments 1: The great effort, dedication, and time invested by the authors in this work, as well as the valuable contribution it aims to make to the field of veterinary radiography, are truly appreciated. While this work provides interesting insights into the use of VHS and VLAS in veterinary radiography, including its evolution over time, in the reviewer’s opinion, the review does not present sufficiently novel or innovative findings to advance current knowledge in the field. Journals of this level prioritize groundbreaking contributions. A more comprehensive and critical evaluation of existing studies, with a clear identification of gaps and future research directions, would strengthen the contribution.
Response 1: We thank this reviewer for the effort and time to evaluate this paper. This is a review paper and brings together information from other studies, therefore, we consider that novel or innovative findings are beyound the scope of this paper type, but rather concentrating a large amount of information in a logical manner for the reader to be able to understand the major idea of the topic. We consider that the use of these measurement and reference values available in the literature could be helpful to the clinician for a better understanding of the value for performing these tests. Moreover, these two measurements benefit from a large number of publications and to date, there is no review paper addressing this topic. We also discussed future perspectives for these methods, such as AI integration, which we believe it is of novelty, since AI is also at its’ begining even in the human medicine field. Furthermore, we made all the effort to include as much of the valuable availabe data in this review and identify the gaps and comment on some particular aspects such as diferences between studies, normal variation and also the ganeralizability, repeatability and reliability aspects.
Reviewer 2 Report
Comments and Suggestions for Authors
The paper is written with appropriate language, with correct and complete bibliographical sources. It follows correct logic.
Despite being a review, it is necessary to indicate (like in the materials and methods) the research strategies, the search time frame, the inclusion and exclusion criteria, the selection method and the type of grouping of publications
Furthermore, it lacks schematization that could make the narrative easier to follow (I'll talk about this later).
For example, in chapter 2., (Measuring method) I recommend classifying the relevant bibliography into three different subchapters: e.g. VHS 2.1, VLAS 2.2 and then the correlations with decubitus, cardiac cycle, respiration and BCS. I recommend making several sub-chapters to make reading easier (2.3....)
The division into two subchapters (VLAS and VHA) is recommended in all chapters (3.,4.,5.)
Lines 82-102: the authors describe 6 different papers on the differences into left (LL) and right (RL) lateral recumbence. Written like this, it becomes difficult to discuss and appreciate the differences. I recommend a table
Chapter 3. The use of VHS and VLAS in clinical practice
Here too we first talk about VHS (table 1) and VLAS (tab 2) measurements in the different breeds. Then the VHS is taken again to talk about the value of serial measurements in the same dog and then the same for the VLAS.
I suggest talking about VHS (3.1) first, differentiating the measurements in the races from the serial ones with different graphics. Same thing for VLAS
Table 1: in the second paper (Sleeper and Buchanan) in the reference values ​​column the authors says MO, not shown in the legend. Specify what it is.
Chapter 4. Vertebral heart score and vertebral left atrial size in heart disease.
It is the part that presents the most confusion, probably due to the many different data, also expressed in different ways (median, minimum-maximum, Q1-Q3, number followed by v, number alone).
The authors need to find a simpler way, also using different graphics, to present the data from lines 188 to 262. The whole part presenting the different VHS results in MMVD dogs (up to line 254) is difficult to follow. I recommend putting everything in the table, both the progression of the disease and the severity of the disease, so that the reader can get an idea with a quick glance, as in table 1 and 2. I would also differentiate the papers by topic , from lines 263 to 294. Same thing for VLAS.
Chapter 5. Repeatability and reproducibility
Same thing: many numbers, not very usable.
Author Response
Comments1: The paper is written with appropriate language, with correct and complete bibliographical sources. It follows correct logic.
Response 1: We would like to thank this reviewer for the availability and time to offer valuable suggestions for improving this manuscript.
Comments 2: Despite being a review, it is necessary to indicate (like in the materials and methods) the research strategies, the search time frame, the inclusion and exclusion criteria, the selection method and the type of grouping of publications
Response 2: We added Section 2 – Materials and methods as following:
Line 66: The search for literature was conducted within the period 1995 and 2024 and in-cluded PubMed and Google Scholar databases. The search in PubMed database was performed by the primary author using the “advanced search” option based on com-bined keywords as following: “((radiogr*) AND ((heart size) OR (cardiac size))) AND ((dog) OR (canine)” resulting in 176 documents, “(((radio*) AND ((cardiac silhouette) OR (heart silhouette))) AND (vertebral)) AND ((dog) OR (canine))” resulting in 21 documents, “((vertebral heart) OR (vertebral left atrial)) AND ((dog) OR (canine))” re-sulting in 187 documents and “(((radio*) AND ((heart) OR (cardiac))) AND ((dog) OR (canine))) AND (((artificial intelligence) OR (deep learning)) OR (machine learning))” resulting in 27 papers. The papers were then screened by the primary author to elimi-nate any articles that did not fulfil the inclusion criteria. From PubMed database, 68 paper were selected in this review. Papers were exported in EndNote v. 21 and dupli-cated documents were manually deleted. Four documents were retrieved from Google Scholar and 13 documents were retrieved from other books and articles bibliography. Articles in English, that focused on veterinary radiographic measurement of heart size in dogs were selected. Articles that were focusing on other species, conference abstracts, and studies lacking full-text availability were excluded. Data from selected studies was extracted, synthetized and organized according to the main subchapters of this review.
Comments 3: Furthermore, it lacks schematization that could make the narrative easier to follow (I'll talk about this later).
For example, in chapter 2., (Measuring method) I recommend classifying the relevant bibliography into three different subchapters: e.g. VHS 2.1, VLAS 2.2 and then the correlations with decubitus, cardiac cycle, respiration and BCS. I recommend making several sub-chapters to make reading easier (2.3....)
Response 3: Changes were performed as suggested
Comments 4: The division into two subchapters (VLAS and VHA) is recommended in all chapters (3.,4.,5.)
Response 4: Changes were performed as suggested
Comments 5: Lines 82-102: the authors describe 6 different papers on the differences into left (LL) and right (RL) lateral recumbence. Written like this, it becomes difficult to discuss and appreciate the differences. I recommend a table
Response 5: We agree that this might be somehow confusing, however we did try to include the information gathered from as many studies as available. In fact, a difference in positioning was found only in 3 studies (references 18, 19 and 20) while other studies did not find any difference (references 6, 21, 13). Moreover, we commented on this aspect as following: Line 134: The effect of dog positioning on VHS remains unclear since there are contradicting re-sults, however, within breed it seems that there is no major influence. However, whenever reference ranges are not available for both left and right lateral position the clinician should take the view as where reference ranges are available.
We consider that adding another table only for 6 references that show different results would complicate even more. Moreover, where available, the reference ranges for LL and RL are presented in table 1 and 2.
To facilitate the reading of this paper we also made the following changes:
We added:
Line 85: 3.1 Vertebral heart score measurement
Line 105: 3.2. Vertebral left atrial measurement
Line 118: 3.3. Measurement variation in normal subjects – and here we moved the discussions with recumbencies, BCS, respiratory phase etc.
Comments 6: Chapter 3. The use of VHS and VLAS in clinical practice
Here too we first talk about VHS (table 1) and VLAS (tab 2) measurements in the different breeds. Then the VHS is taken again to talk about the value of serial measurements in the same dog and then the same for the VLAS.
I suggest talking about VHS (3.1) first, differentiating the measurements in the races from the serial ones with different graphics. Same thing for VLAS
Response 6: Changes were performed as suggested:
Therefore, we added:
Line 171: 4.1 The use of reference ranges for VHS and VLAS
Line 191: 4.2 The use of serial measurements for VHS and VLAS in the disease progression
Comments 7: Table 1: in the second paper (Sleeper and Buchanan) in the reference values ​​column the authors says MO, not shown in the legend. Specify what it is.
Response 7: Corrected as suggested
Comments 8: Chapter 4. Vertebral heart score and vertebral left atrial size in heart disease.
It is the part that presents the most confusion, probably due to the many different data, also expressed in different ways (median, minimum-maximum, Q1-Q3, number followed by v, number alone).
The authors need to find a simpler way, also using different graphics, to present the data from lines 188 to 262. The whole part presenting the different VHS results in MMVD dogs (up to line 254) is difficult to follow. I recommend putting everything in the table, both the progression of the disease and the severity of the disease, so that the reader can get an idea with a quick glance, as in table 1 and 2. I would also differentiate the papers by topic , from lines 263 to 294. Same thing for VLAS.
Response 8: We totally agree with this reviewer regarding the multitude of data and various expression of the results. However, besides the general rule or presentation based on data distribution (mean and standard deviation or median and interquartile range), there is no consensus of presenting the data in the literature. Some studies present their data as mean and standard deviation, median and IQR, 95%CI or range. In this type of study, one can simply condense these data and draw some conclusions based on the "general" output resulted from this analysis. However, to facilitate the reader’s perception on this large amount of data available, as suggested, we created graphic bars representing the median and range (minimum and maximum) values for both VHS and VLAS measurements in different classess of MMVD. We also created a plot that includes the cutoff value for cardiomegaly in MMVD based on different studies from the literature: Figure 1, 2, 3 and 4.
Comments 9: Chapter 5. Repeatability and reproducibility
Same thing: many numbers, not very usable.
Response 9: As suggested, we added a table (Table 3) with the ICC value for inter- and intra-observer correlation coefficient base on multiple studies.
Reviewer 3 Report
Comments and Suggestions for Authors
Dear colleagues,
I would like to congratulate you on the writing of this manuscript. This is a very thorough revision of the literature related to a very popular subject among veterinarians, radiographic cardiac silhouette measuring. Your work passes through almost all the papers that have been published about VHS and VLAS so far. And the information gathered from that reviewing job is presented in an organized manner, concise and focused on the relevant information. Basically, the cutoffs and their sensitivity and specificity, which in the end is what users of the methods rely on. But, as you point out wisely, any measurements should be interpreted along the clinical presentation.
The adding of the current situation of radiographic interpretation regarding AI is concerned is very interesting. In humane as in veterinary medicine this is where diagnostic imaging is moving to, and it is a field yet to explore.
Only a writing mistake found on Line 370, a "was" after VHS should be deleted.
Thank you for your work.
Author Response
Dear colleagues,
Comments 1: I would like to congratulate you on the writing of this manuscript. This is a very thorough revision of the literature related to a very popular subject among veterinarians, radiographic cardiac silhouette measuring. Your work passes through almost all the papers that have been published about VHS and VLAS so far. And the information gathered from that reviewing job is presented in an organized manner, concise and focused on the relevant information. Basically, the cutoffs and their sensitivity and specificity, which in the end is what users of the methods rely on. But, as you point out wisely, any measurements should be interpreted along the clinical presentation.
The adding of the current situation of radiographic interpretation regarding AI is concerned is very interesting. In humane as in veterinary medicine this is where diagnostic imaging is moving to, and it is a field yet to explore.
Response 1: We would like to thank this reviewer for the effort to evaluate and improve our paper.
Comments 2: Only a writing mistake found on Line 370, a "was" after VHS should be deleted.
Response 2: Corrected as suggested.
Thank you for your work.

Reviewer 4 Report
Comments and Suggestions for Authors
Although echocardiography is currently most commonly used to assess heart size, vertebral heart score is still used in the assessment of cardiomegaly. A solid review of the literature on vertebral heart score and vertebral left atrial size. The authors have reviewed many studies and summarised them in a logical manner. A particularly interesting part is the use of VHS and VLAS in various heart diseases which may contribute to numerous citations of this article.
Author Response
Comments 1: Although echocardiography is currently most commonly used to assess heart size, vertebral heart score is still used in the assessment of cardiomegaly. A solid review of the literature on vertebral heart score and vertebral left atrial size. The authors have reviewed many studies and summarised them in a logical manner. A particularly interesting part is the use of VHS and VLAS in various heart diseases which may contribute to numerous citations of this article.
Response 1: We thank this reviewer for the effort and availability to evaluate this paper.

Round 2
Reviewer 1 Report
Comments and Suggestions for Authors
I appreciate the authors’ efforts in addressing the previous comments. However, after careful evaluation, I find that the manuscript still does not meet the minimum scientific and methodological standards required for publication in this journal.
Reviewer 2 Report
Comments and Suggestions for Authors
The authors responded adequately and clearly to my comments; now the work is complete and easily understandable, despite the numerous data reported